

# Latent heating profiles from GOES-16 and its comparison to heating from NEXRAD and GPM

Yoonjin Lee[1], Christian D. Kummerow[1,2], Milija Zupanski[2]

[1]Department of Atmospheric Science, Colorado State University, Fort Collins, Colorado, 80521, USA
[2]Cooperative Institute for Research in the Atmosphere, Fort Collins, Colorado, 80521, USA

*Correspondence to*: Yoonjin Lee (yoonjin.lee@colostate.edu)

**Abstract.** Latent heating (LH) is an important quantity in both weather forecasting and climate analysis, being the essential factor driving convective systems. Yet, inferring LH rates from our current observing systems is challenging at best. For climate studies, LH has been retrieved from the Precipitation Radar (PR) on the Tropical Rainfall Measuring Mission (TRMM) using
model simulations in the look-up table (LUT) that relates instantaneous radar profiles to corresponding heating profiles. These radars, first on TRMM and then Global Precipitation Measurement (GPM), provide a continuous record of LH. However, with observations approximately 3 days apart, its temporal resolution is too coarse to be used to initiate convection in forecast models. In operational forecast models such as High-Resolution Rapid Refresh (HRRR), convection is initiated from LH derived from ground based radar. Despite the high spatial and temporal resolution of ground-based radars, one disadvantage of using it is that
its data are only available over well observed land areas. This study suggests a method to derive LH from the Geostationary Operational-Environmental Satellite-16 (GOES-16) in near-real time. Even though the visible and infrared channels on the Advanced Baseline Imager (ABI) provide mostly cloud top information, rapid changes in cloud top visible and infrared properties, when coupled to a LUT similar to those used by the TRMM and GPM radars, can equally be used to derive LH profiles for convective regions using model simulations coupled to a convective classification scheme and channel 14 (11.2 μm)
brightness temperature. Convective regions detected by GOES-16 are assigned LH from the LUT, and they are compared with LH from NEXRAD and one of Dual-frequency Precipitation Radar (DPR) products, Goddard Convective-Stratiform Heating (CSH). LH obtained from GOES-16 show similar magnitude with NEXRAD and CSH, and vertical distribution of LH is also very similar with CSH. Overall, GOES LH appear to have the ability to mimic LH from radars, although the area identified as convective is roughly 25% smaller than the current HRRR model, while the heating is correspondingly higher.

**1 Introduction**

As the spatial resolution of numerical weather prediction models becomes finer, and even operational models are run at resolutions of a few kilometer, an effective way to assimilate observation data at this fine resolution has been sought (Gustafsson et al., 2018). At a few kilometer resolution, convection can be resolved explicitly (Seity et al., 2011). However, if the model environment is not favorable for convection, updrafts and clouds will not develop in the right place. In order to correctly initiate
convection in operational regional models where both accuracy and speed are fairly important, observed latent heating (LH) can be added in the model in the data assimilation cycle. LH is not only important to initiate convection, it also contributes to the intensification of convection. Adding LH increases buoyancy in the atmospheric column, thereby inducing convection, and it has become an important procedure that many operational models use for the initialization of convective events (Gustafsson et al., 2018).




National Oceanic and Atmospheric Administration (NOAA)'s operational models, the Rapid Refresh (RAP) and High-Resolution Rapid Refresh (HRRR), both use observed latent heating to drive convection, but in different ways (Benjamin et al., 2016). RAP uses digital-filter initialization (Peckham et al., 2016) while HRRR simply replaces modeled temperature tendency with the observed LH (Benjamin et al., 2016). For this operational purpose, LH data have to be available continuously in near-

real time. Therefore, ground-based radars which have high spatial and temporal resolutions similar to HRRR's resolutions are used to calculate LH from reflectivity in HRRR. While suitable for the HRRR region of interest, the method is not applicable to regions beyond radar coverage such as the Gulf of Mexico and even some mountainous areas. It also limits the model's applicability to global scales.

The Precipitation Radar (PR) on the Tropical Rainfall Measuring Mission (TRMM) has been used to retrieve LH from space. PR is the first meteorological radar in space, designed to provide vertical distribution of precipitation over the tropics (Kummerow et al., 1998). From its three-dimensional precipitation data, vertical profiles of LH have been retrieved. There are several retrieval algorithms using PR: Goddard Convective-Stratiform heating (CSH; Tao et al., 1993), Spectral Latent Heating (SLH; Shige et al., 2004), Hydrometeor heating (HH; Yang and Smith, 1999), and Precipitation Radar Heating algorithm (PRH; Satoh and

Noda, 2001). Among these algorithms, CSH and SLH are the two most widely used products. Most recent versions of monthly gridded CSH and SLH products have spatial resolution of 0.25°×0.25° and 0.5°×0.5° respectively with 80 vertical layers and have been used to provide valuable insights to heat budget or atmospheric dynamics over Tropics. Two algorithms have improved since their first development, and both algorithms are also applied to Dual-frequency Precipitation Radar (DPR) data on Global Precipitation Measurement (GPM), the successor of TRMM, to continue the climate record of LH and expand the

regions of interest to mid-latitude.

CSH and SLH both rely on the lookup table (LUT) based on cloud resolving model simulations. Inputs that are used to look for LH profiles in the LUT are different, but their common inputs to the LUT are echo top height and surface rainfall rate. Echo top height is important in determining the depth of heating in the vertical, and surface rainfall rate is a good indicator for intensity of

maximum heating rate. Even though the methods use different model simulations to create the LUT, and differ in other details, they seem to exhibit similar distributions when they are averaged spatially or temporally (Tao et al., 2016).

Although these products are considered instantaneous heating, their temporal resolutions are poor compared to 15-minute or hourly observations available from ground-based radars. Geostationary data is required to achieve the sampling of ground-based

radars. The visible (VIS) and infrared (IR) sensor on geostationary satellite, unfortunately, cannot provide much vertical information as active sensor do in the presence of thick clouds, but their data contain cloud top information, and rapid refresh provides important information about a cloud's convective nature. Cloud top information from geostationary data is included when creating cloud analysis during data assimilation (Benjamin et al., 2016), and thus LH retrieved based on cloud top temperature, can be useful in the forecast model by keeping consistency of retrieved LH with the updated cloud analysis.


This study examines if cloud convective identification from the cloud's one-minute evolution from the Advanced Baseline Imager (ABI), coupled with cloud top information on Geostationary Operational-Environmental Satellite-16 (GOES-16), can be sufficient to approximate NEXRAD-derived LH. Following the lead of spaceborne radar LH algorithms, a LUT is created using model simulations. Once convective clouds are determined by using 10 consecutive one-minute ABI data, LH profiles for

convective clouds are looked for in the LUT based on cloud top temperature of the convective cloud. Unlike DPR products that



has temporal resolution of a day, ABI data in mesoscale sector are provided with one-minute interval, and thus LH can be obtained from GOES-16 as frequently as NEXRAD, thereby eligible for initiating convection during the forecast. LH from GOES-16 can be beneficial over the regions without radar such as ocean or mountainous regions where the quality of radar data degrades.


Detailed descriptions of CSH, SLH and how NEXRAD converts reflectivity to LH are provided, followed by the retrieval process using GOES-16 ABI. LH from GOES-16 are then examined to see if it is comparable to other radar products.

## 2 Existing LH retrieval methods

### 2.1 Radiosonde networks

LH is not a measurable quantity as it is almost impossible to single out temperature changes by phase changes from the total observed temperature changes. However, heat and moisture budget studies are conducted using sounding network in a field campaign, and apparent heat source ($Q_1$) and apparent moisture sink ($Q_2$) from the budget study are related to LH (Yanai et al., 1973; Johnson 1984; Demott 1996). It can be done using a diagnostic heat budget method which is first presented by Yanai et al. 1973 (Tao et al., 2006). Over a certain horizontal area, $Q_1$ can be expressed as the equation below that includes LH (Tao et al.,

90   2006).

$$Q_1 - Q_2 = \bar{\pi}\left[-\frac{1}{\bar{\rho}}\left(\overline{\frac{\partial \bar{\rho}w'\theta'}{\partial z}}\right) - \overline{\nabla \cdot V'\theta'}\right] + \frac{1}{c_p}\left[L_v(c-e) + L_f(f-m) + L_s(d-s)\right] \tag{1}$$

where prime denotes deviations from horizontal averages, which is denoted by upper bar. $Q_R$ is the radiative heating rate, $\theta$ is potential temperature, $\pi$ is non-dimensional pressure, $\rho$ is air density, $c_p$ is specific heat at constant pressure and R is gas constant for dry air. $L_v$, $L_f$, and $L_s$ represent the latent heats of condensation, freezing, and sublimation while c, e, f, m, d, and s represent

each microphysical process of condensation, evaporation, freezing, melting, deposition, and sublimation, respectively. The last six terms on the right-hand side that include these microphysical processes are LH from phase changes. Since $Q_1$ can be obtained using vertical profiles of temperature, moisture, and wind data observed during the field campaign (Tao et al., 2006), the observed $Q_1$ is used to indirectly validate GPM LH products that are retrieved together with $Q_1$.

### 2.2 CSH and SLH from GPM DPR

DPR has two operational LH algorithms: the Goddard Convective-Stratiform Heating (CSH) and Spectral Latent Heating (SLH). In the GPM products, LH is provided along with additional variables: $Q_1$-$Q_r$ and $Q_2$ in SLH and $Q_1$-$Q_r$-LH, $Q_r$, and $Q_2$ in CSH (Tao et al., 2019). These algorithms are first developed for TRMM data, but have been adapted to GPM data. Both algorithms use cloud resolving model simulations to create a LUT relating hydrometeor profiles to cloud model heating rates. Although there is no direct measurement for LH to validate the results, retrieved $Q_1$ and $Q_2$ are compared instead with sounding data from

various field campaigns through a method mentioned in section 2.1. The evolution of these products is well summarized in (Levizzani et al., 2020), but each algorithm is briefly explained here.

The CSH algorithm is first introduced by Tao et al. 1993. Initial algorithm by Tao et al.1993 uses surface rainfall rate and amount of stratiform rain as inputs to the LUT, but the LUT has been improved by increasing the number of LH profiles, using

finer resolution in simulations, and adding new inputs such as echo-top heights and low-level vertical reflectivity gradients (Tao et al., 2019). For high-latitude regions observed by GPM satellite, new LUT is created with simulations from NASA Unified-



Weather Research and Forecasting model which is known to be suitable for high latitude weather system (Levizzani et al., 2020). Inputs to this LUT are surface rainfall rate, maximum reflectivity height, freezing level height, echo top height, decreasing flag, and maximum reflectivity intensity (Levizzani et al., 2020).


The SLH algorithm is based on Shige et al. 2004 and Shige et al. 2007. For tropical regions, the LUT is created for three different rain types; convective, shallow stratiform, and anvil (or deep stratiform) clouds. Inputs to the LUT are precipitation top height (PTH), precipitation rate at the surface ($P_s$), precipitation rate at the level that separates upper-level heating and lower-level heating ($P_f$) and precipitation at the melting level ($P_m$). Once non-convective rain is separated into either shallow stratiform

or anvil, a vertical profile for anvil cloud is chosen based on $P_m$, and magnitudes of upper level heating and lower level cooling are normalized by $P_m$ and ($P_m - P_s$), respectively. For convective and shallow stratiform clouds, a vertical profile corresponding to the PTH is chosen, and then upper-level heating and lower-level heating are normalized by $P_f$ and $P_s$, respectively. For DPR, a new LUT is created for mid and higher latitude to account for expanded latitudinal coverage by GPM. For higher latitude regions, six precipitation types (convective, shallow stratiform, three types of deep stratiform, and other) instead of three are

used, and therefore six respective LUTs exist. Inputs to these LUTs are precipitation type, PTH, precipitation bottom height, maximum precipitation, and $P_s$.

Figure 1 shows monthly gridded products from these two algorithms over CONUS for July of 2020 at four different heights. Overall horizontal pattern in the two products looks similar. However, there is a difference in the vertical. At 5km or 8km, CSH

tends to show higher heating rate especially over mid-latitude, while at 10km, SLH shows higher heating rate. In addition, SLH tends to have larger cooling rate throughout the layers, although it is not clear from the figure. These discrepancies would be attributed to different configuration setup such as microphysical scheme used to run simulations for the LUT. This again shows that there is no true heating rate that we can trust, and vertical profiles of LH highly depend on the simulations that comprise the LUT.













**Figure 1: Monthly gridded LH from CSH at (a) 2km, (c) 5km, (e) 8km, and (g) 10km and LH from SLH at (b) 2km, (d) 5km, (f) 8km, and (h) 10km.**




Orbital data for these products have finer spatial resolution of 5km, but their temporal resolution is too coarse to be used in the forecast model, which typically has a spatial resolution of few kilometers and time step of few seconds. The closest that can meet the resolutions of the forecast model is ground-based radar data, and this is the reason why LH derived from ground-based radar is used to initiate convection during the short-term forecast.

### 2.3 LH from NEXRAD

In the operational forecast model, LH profiles retrieved using radar reflectivity replace modeled LH profiles so that appropriate heating rate can help initiate convection. LH profiles in this case are obtained through a simple empirical formula that converts radar reflectivity to LH. In Eq. (2), reflectivity is converted to potential temperature tendency using model pressure field. This equation is only applied when radar reflectivity exceeds 28dBZ. The threshold of 28dBZ was chosen based on the effectiveness of adding heating from reflectivity in HRRR (Bytheway et al., 2017).

$$T_{ten} = \frac{1000^{R_d/c_{pd}}}{p} \frac{(L_v + L_f)Q_s}{n \cdot c_{pd}} \tag{2}$$

where $Q_s = 1.5 \times \frac{10^{z/17.8}}{264083}$

z: grid radar/lightning-proxy reflectivity

Tten: temperature tendency

p: background pressure (hPa)

$R_d$: specific gas constant for dry air

$c_{pd}$: specific heat of dry air at constant pressure

$L_v$: latent heat of vaporization at 0°C

$L_f$: latent heat of fusion at 0°C

N: number of forward integration steps of digital filter initialization

$T_{ten}$ in Eq. (2) is produced in K/s to meet the needs during the short-term forecast. Although heating rate is not a general output in the forecast model, it is calculated every time step by dividing temperature change from microphysical scheme by time step which is usually on the order of few tens of seconds. Therefore, this empirical formula is developed to produce LH comparable to the modeled heating rate in K/s so that added LH does not blow up the model when it is ingested.

### 3 LH profiles from GOES-16

Current operational geostationary satellite, GOES-16, carries the Advanced Baseline Imager (ABI), an instrument with 16 VIS and IR channels. Mesoscale sectors, which are manually moved around to observe interesting weather events, provide data in one-minute intervals. Such high temporal resolution data have helped observe cloud developments in more detail. Using this high temporal resolution ABI data, convective clouds are detected, and LH profiles for the detected clouds are assigned from a lookup table. The lookup table is created running the Weather Research and Forecasting (WRF) model simulations. While CSH and SLH algorithm look for LH profiles in a model-based LUT according to precipitation type and precipitation top height, the LUT for GOES is created for convective clouds that appear bright and bubbling from ABI according to brightness temperature ($T_b$) at channel 14 (11.2μm), which is a good indicator of cloud top temperature. LH is not assigned for stratiform clouds from GOES-16 as it is not important in initiating convection in the forecast. Once convective clouds are detected using temporal changes in reflectance and $T_b$, LH profile corresponding to the $T_b$ of the detected cloud is assigned from the LUT.



### 3.1 Definition of convection in GOES-16 ABI and model simulations

When using LH to drive convection in the operational forecast model, LH derived from radar reflectivity is applied only in convective regions. In HRRR, a simple threshold of 28dBZ determines where to put LH, but there is no such simple threshold for

VIS or IR channels that can determine convective regions. However, there are several convection detecting algorithms for GOES-16 ABI, including Lee et al. 2020. It uses mesoscale sector data with one-minute interval to detect convections from ABI. Two separate detecting methods are proposed for vertically growing clouds in early stages and mature convective clouds that move rather horizontally once it reaches the tropopause and often have overshooting tops. The method for vertically growing clouds measures $T_b$ decrease over ten minutes for two water vapor channels, and if the decrease is greater than the designated

threshold, it assigns the pixel as convective. For mature convective clouds, the method looks for grid points that have continuously high reflectance, low $T_b$, and lumpy cloud top over ten minutes. Combining the two methods provides results comparable to radar product, and these methods are rather simple and fast. Therefore, this algorithm is used to detect convective regions from ABI.

However, this method is not applicable to model simulations due to unreliable reflectance simulated by the Community Radiative Transfer Model (CRTM). Instead, convection is defined with vertical velocity, which is one of prognostic variables in the model. It is actually the most direct and accurate way to define convection (Zipser & Lutz, 1994; LeMone &Zipser, 1980; Xu & Randall, 2001; Wu et al., 2009; Delgenio et al., 2010; Schumacher et al., 2015), but not widely used since vertical velocity is not always available in observation data. A threshold is usually defined at a certain altitude or over certain range of altitudes for a

general use. However, vertical velocity tends to peak at different height at different stages of convection (Schumacher et al. 2015), and not one altitude works for all the convection. Therefore, an appropriate threshold for the model simulation that is also consistent with the observed scene is determined in this study, not pursuing values from previous studies.

$T_b$ at 11.2μm which is used to construct the LUT is mostly sensitive to hydrometeors or water vapor. Accordingly, the signal

received by the channel will be largely from layers with high cloud water contents. Considering that cloud water is produced after an updraft followed by condensation, an altitude that has maximum cloud water contents can be regarded as an altitude with the strongest updraft. Since vertical velocity at a layer with maximum cloud water contents can be beneficial in both determining convection at all stages and matching with the observation, it is used in this study with a threshold that can keep consistency between model outputs and observation. The threshold is chosen comparing fractions of convective regions. Table 1 shows

convective fractions using the GOES-16 convection detecting algorithm and using different vertical velocity thresholds in the model outputs. Using higher thresholds can prevent including non-convective grids, but at the same time, it will only include the strongest part of convective regions. Using 1.5m/s shows a fraction closest to the observed fraction, and therefore, 1.5m/s is used to define convection in the model output. This number is actually similar to values used in some previous studies (1m/s in LeMone and Zipser 1980, Xu and Randall 2001, and Wu et al., 2009)


**Table 1.** Fraction of convective regions in observation and using different vertical velocity thresholds in the model output.

| Observation | 1m/s | 1.5m/s | 2m/s | 3m/s | 4m/s |
| --- | --- | --- | --- | --- | --- |
| 0.96% | 1.53% | 1.04% | 0.77% | 0.47% | 0.31% |


### 3.2 Model simulations used to create a lookup table

11 convective cases are simulated using WRF to obtain stable mean LH profiles. The convective cases are chosen over CONUS
within NEXRAD network during May to August in 2017 or 2018. All simulations use the same configuration in Table 2 to avoid
discrepancy between simulation results, and their $T_b$s at 11.2μm are calculated using the CRTM. In each scene, convective grid
points are defined by the threshold found in the previous section, and neighboring convective grid points are clustered to form a
convective cloud. Minimum $T_b$ of each cloud is calculated, and LH profiles from the clouds with the same minimum $T_b$ are
averaged to produce mean profiles for each $T_b$ bin of the LUT. LH profiles gathered in the LUT are provided in K/s as for
NEXRAD.

**Table 2.** Table for WRF simulation setup.

| Version | WRFv3.9 |
| --- | --- |
| Spatial resolution | 3km |
| Time step | 10 seconds |
| Microphysical scheme | Aerosol-aware Thompson scheme (The original scheme is modified to produce vertical profiles of LH as outputs) |
| Planetary boundary layer | Mellor-Yamada Nakanishi Niino (MYNN) Level 2.5 and Level 3 schemes |
| Land surface model | Rapid update cycle (RUC) land surface model |
| Long wave and short wave radiation physics | Rapid radiative transfer model for general circulation models (RRTMG) schemes |

### 3.3 Mean LH profiles according to cloud top temperature

LH profiles of convective clouds from 11 WRF simulations are collected according to 16 bins of the minimum cloud top
temperature at 11.2μm. 16 bins range from below 200K to above 270K with a bin size of 5K. Figure 2 shows mean vertical
profiles of LH in each bin. All profiles exhibit slightly negative LH near the ground due to evaporation, but positive LH is shown
at most layers. It is also nicely shown in the figure that as the $T_b$ decreases, the profile stretches up in the vertical. Interestingly
though, the maximum heating rate is not perfectly proportional to $T_b$. However, considering the maximum LH that is allowed in
HRRR model, which is 0.02K/s, these values seem quite reasonable. Table 3 shows average of maximum surface precipitation
rate and mean surface precipitation rate for each bin. Precipitation rate is mostly inversely proportional to $T_b$ in Table 3. This is
expected as deeper and higher clouds tend to precipitate more. This again shows that mean LH profiles for each bin are properly
obtained from GOES-16.










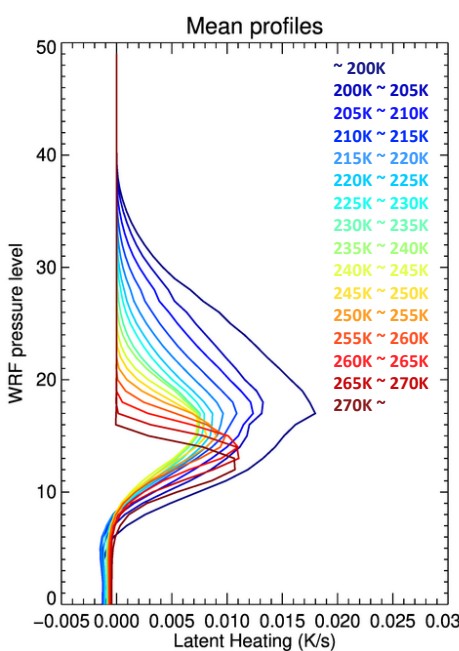

**Figure 2: Mean vertical profiles for each cloud top temperature bin.**

**Table 3.** Table of maximum precipitation rate and mean precipitation rate for each cloud top temperature bin.

|  | Maximum precipitation rate (mm/hour) | Mean precipitation rate (mm/hour) |
|---|---|---|
| ~200K | 137.4 | 48.8 |
| 200K ~ 205K | 99.7 | 41.2 |
| 205K ~ 210K | 88.0 | 47.2 |
| 210K ~ 215K | 60.9 | 40.0 |
| 215K ~ 220K | 41.6 | 30.3 |
| 220K ~ 225K | 31.1 | 23.5 |
| 225K ~ 230K | 24.9 | 18.9 |
| 230K ~ 235K | 20.1 | 15.5 |
| 235K ~ 240K | 16.4 | 12.6 |
| 240K ~ 245K | 14.0 | 13.4 |
| 245K ~ 250K | 10.8 | 10.9 |
| 250K ~ 255K | 10.4 | 10.9 |
| 255K ~ 260K | 7.9 | 7.4 |
| 260K ~ 265K | 6.4 | 6.0 |
| 265K ~ 270K | 4.8 | 4.1 |
| 270K ~ | 3.4 | 3.1 |






## 4 Comparisons between products

LH from three different instruments, GOES-16, NEXRAD, and DPR are examined for comparison. Methods using GOES-16 and DPR products are similar in a way that they use cloud top height or PTH to look for mean profiles in the LUT created with model simulations, although DPR has additional parameters such as surface rain rate which is used to vary the magnitude of the heating rate. In contrast, NEXRAD uses an empirical formula to convert radar reflectivity to LH regardless of PTH. They are all instantaneous heating, but provided in different units. LH from GOES-16 and NEXRAD is in K/s to easily match with modeled heating rate, while DPR products are in K/hour. Therefore, LH in K/hour from DPR products are converted to K/s for comparison.

A scene on 18th June, 2019 is shown in Fig, 3 to compare how each product determines precipitation type (convective or stratiform) which is one of the major factors in estimating LH profiles. The regions with reflectivity greater than 28dBZ in Fig. 3a are regions where LH is estimated from NEXRAD reflectivity to be used in HRRR, but not necessarily convective regions. These regions are larger than convective regions defined by DPR products in Fig. 3c and include some of the stratiform regions assigned by DPR. Pink regions on top of visible image at channel 2 (0.65µm) in Fig. 3b are convective regions detected by GOES-16, and they encompass smallest regions compared to others. Even though areal coverage differs by the method, locations of convective core matches well between the products.

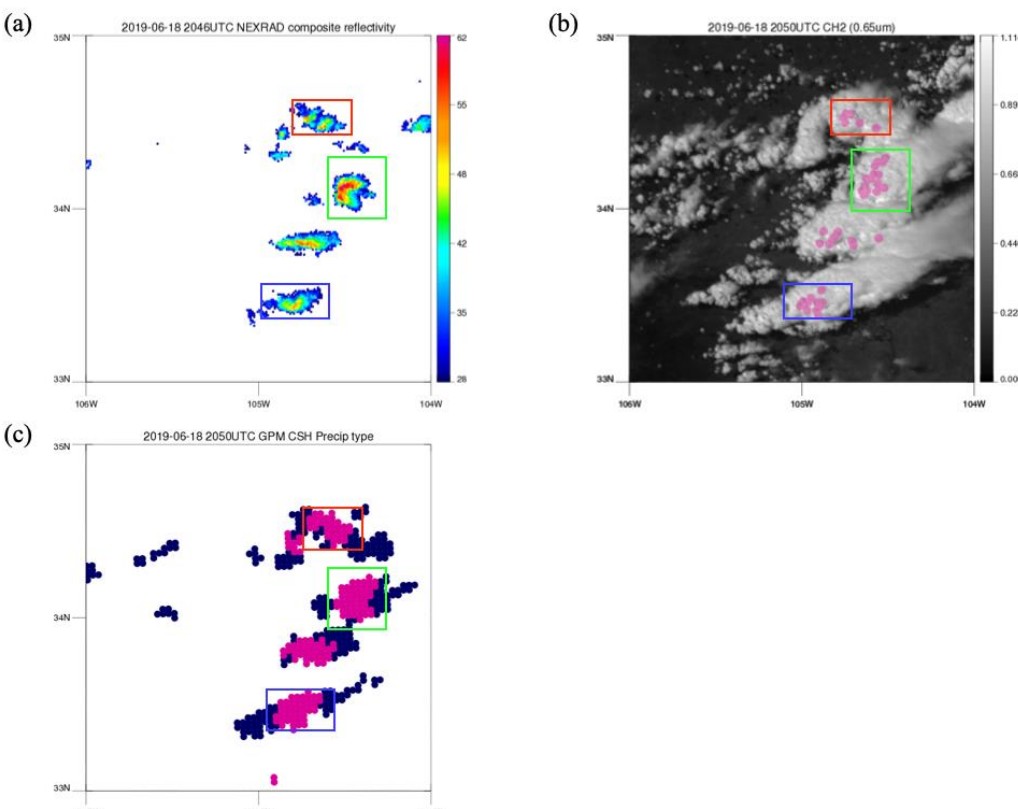

**Figure 3: A scene on June 18th, 2019. (a) NEXRAD composite reflectivity. Only the regions with reflectivity greater than 28dBZ are shown with colors. Color bar is in dBZ. (b) Convective regions detected by GOES-16 are colored in pink on top of GOES-16 visible image at channel 2 (0.65µm). (c) Precipitation type defined by CSH. Convective regions are colored in pink while stratiform regions are colored in navy.**



Clouds in colored boxes in Fig. 3 are all convective clouds, but in different evolutional stages. Clouds in red, green, and blue box respectively have high, low, and mid-level cloud top temperature. LH profiles from NEXRAD, GOES-16, and CSH for these clouds are interpolated into the same WRF grid with 3km resolution for comparison in Figs. 4, 5, and 6. CSH provides LH for both convective and stratiform regions, and thus different colors of lines in Figs. 4c, 5c, and 6c represent different cloud type. Lines with light blue color are each LH profiles of convective grid points in the red box, and blue line is the mean of these

profiles. Similarly, LH profiles of each stratiform gird point are in light green, and the mean of these profiles is in dark green. The total mean LH profile is colored in red. Convective LH profiles from CSH shows heating throughout the vertical layers as expected, except near the surface due to evaporation at lower levels. LH profiles in stratiform regions show cooling at low levels below a melting level and heating above. LH profiles from GOES-16 corresponding to the three convective clouds are shown in Figs. 4b, 5b, and 6b, light blue line being each profile and blue line representing the mean. Even though mean profile is assigned

from GOES-16 for each convective cloud, a number of different lines are shown in the figure due to spatial interpolation. When LH profiles from GOES-16 and CSH are compared, mean profile of convective LH from CSH in blue (Figs. 4c, 5c, and 6c) is similar to GOES LH in blue (Figs. 4b, 5b, and 6b) both in terms of the magnitude and the vertical shape.

On the other hand, LH from NEXRAD shows different vertical shape from GOES-16 or CSH which uses the LUT consisting of

model simulations. LH profiles from GOES-16 or CSH peak around the middle of the atmosphere while NEXRAD LH in convective core (Figs. 4a, 5a, and 6a) tends to peak at low levels where radar reflectivity is high. At low levels where model simulations have cooling, NEXRAD LH does not show cooling due to Eq. (2) which is designed to only produce positive values. This heating at lower levels can help increase buoyancy in lower atmosphere, and thus, convection can be effectively initiated from the added heating.


Although their vertical shape is different, the magnitude of LH is within similar magnitude. Overall values of mean LH profile from NEXRAD in blue is slightly smaller than mean profile from GOES-16 or mean convective LH profile from CSH (blue line), but are closer to the total mean profile of CSH (red line), which indicates that 28dBZ threshold might include some stratiform regions as well. Smaller mean of NEXRAD LH is mainly attributed to anvil regions where reflectivity greater than

28dBZ only exist at few vertical layers and 0dBZ elsewhere.








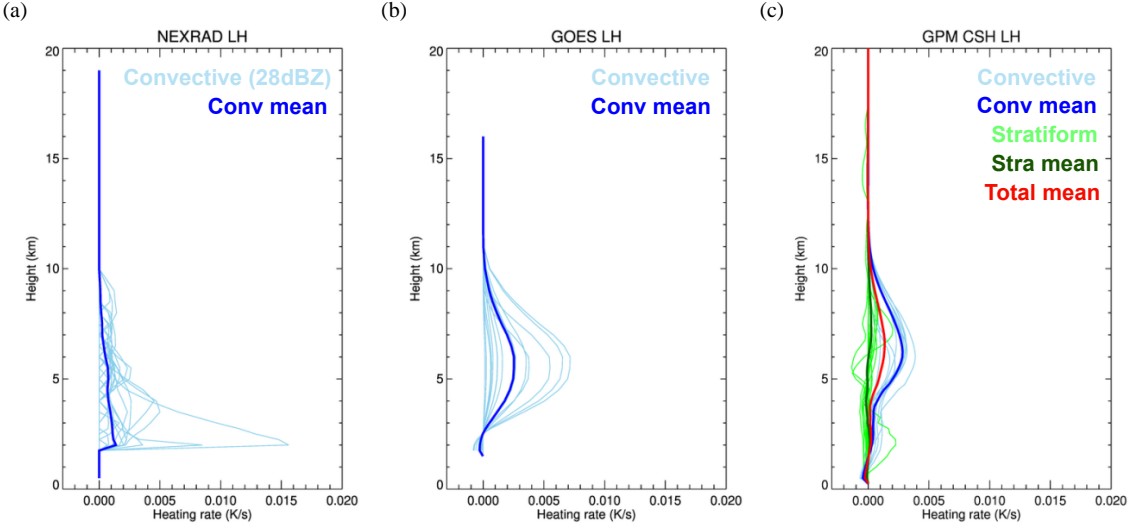

**Figure 4:** LH profiles from (a) NEXRAD, (b) GOES-16, and (c) CSH for the red box region. Light blue lines are each LH profile for convective grid point and blue line is a mean profile of the light blue lines. In (c), each LH profile for stratiform grid point is coloered in light green and its mean profile is colored in dark green. The total mean of LH profiles for CSH is colored in red.

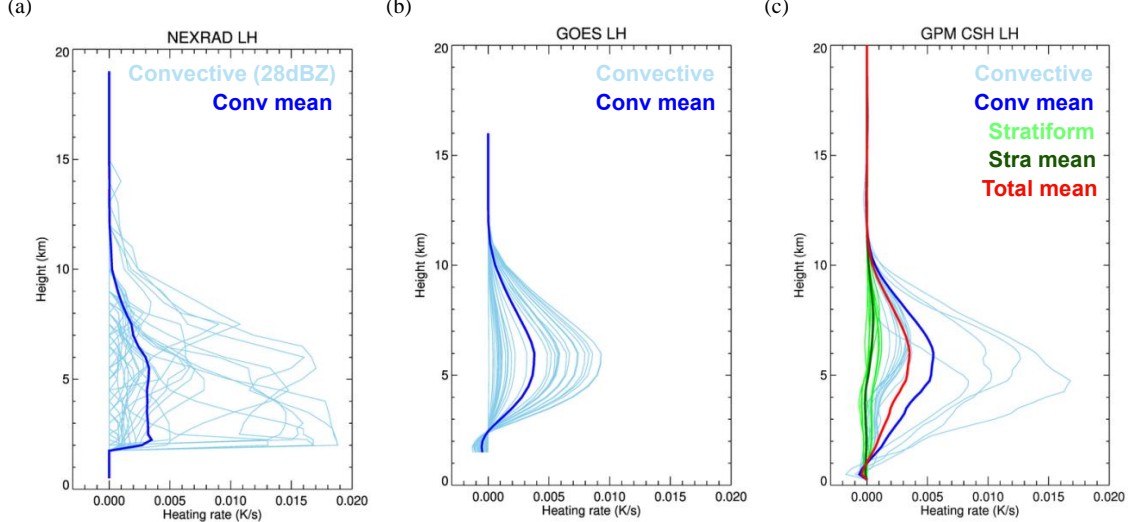


**Figure 5:** Same as Fig. 4, but for the green box region.






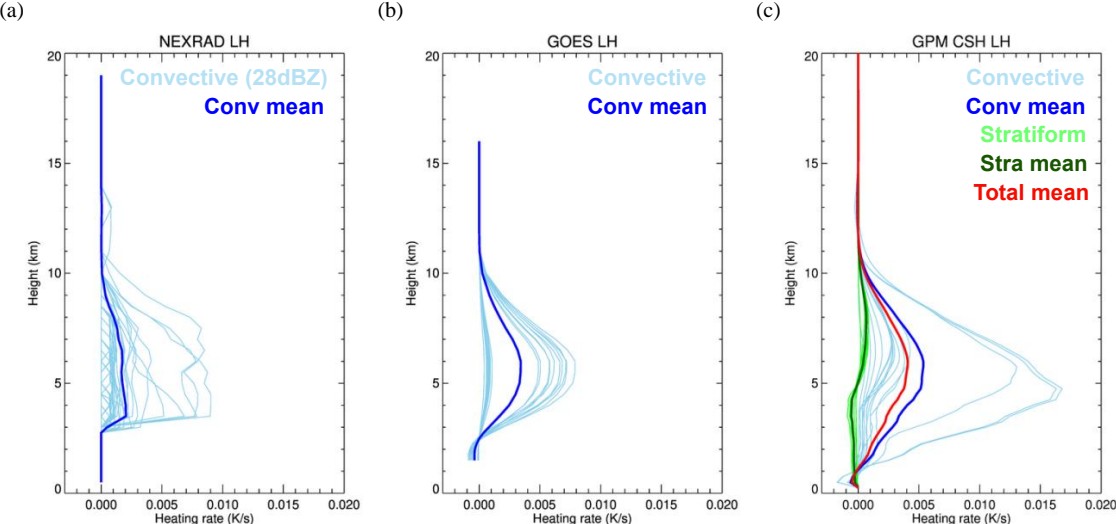

**Figure 6: Same as Fig. 4, but for the blue box region.**

Even though the mean LH from NEXRAD is smaller, the total LH for the region can be similar when it is added up over the region due to broader area determined by the threshold of 28dBZ in Fig. 3a than GOES-16 detection in Fig. 3b. Therefore, the total LH of each cloud is compared between GOES-16 and NEXRAD. Figure 7 shows vertical profiles of LH that is horizontally summed over each convective cloud, each color representing colors of the three box regions. As mentioned before, the altitude that LH peaks is different, but the magnitude of the total heating is very similar. Finally, the total LH of each region is obtained

by summing up the vertical profiles in Fig. 7 and presented in Table 4. The total LH is shown to be similar between NEXRAD and GOES-16. Despite the smaller mean LH from NEXRAD that was shown in Figs. 4, 5, and 6, it shows a good agreement in total heating between GOES-16 and NEXRAD.

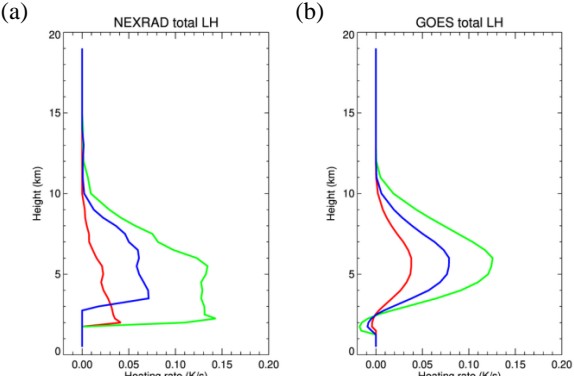

**Figure 7: Vertical profiles of the total heating in the boxed regions from (a) NEXRAD and (b) GOES-16. Different colors represent the color of the box region.**

**Table 4.** Total LH (K/s) from NEXRAD and GOES-16 in the red, green, and blue box regions.

|         | Red  | Green | Blue |
|---------|------|-------|------|
| NEXRAD  | 0.31 | 1.41  | 0.68 |
| GOES-16 | 0.34 | 1.83  | 0.64 |



**5 Conclusions**

A method to obtain vertical profiles of LH from GOES-16 ABI data was described. Convective clouds are first detected using temporal changes in reflectance and $T_b$, and LH profiles for the detected cloud is found by searching a LUT created using WRF

model simulations. The LUT contains LH profiles of convective clouds that are defined by a threshold of 1.5m/s for the modeled vertical velocity, and these convective LH profiles are sorted according to $T_b$ at 11.2μm, which is a good indicator of cloud top height. Mean profiles that represent each $T_b$ bin show good correlation with cloud top temperature, with lower $T_b$ bin having deeper LH profiles. Precipitation rates corresponding to each bin are also well correlated to $T_b$. In addition, maximum LH in the LUT is well within the range that is allowed in HRRR to initiate convection using NEXRAD.


Even though this method is not designed for assigning LH profiles at each grid point as radar products do, it can assign mean values for each cloud. Since the convection detection method for GOES-16 tends to detect convective core region, each cloud is defined separately by combining neighboring grid points, and mean LH is assigned for the cloud. LH from GOES-16, NEXRAD, and CSH are compared in three convective clouds with different cloud top heights. Vertical profiles of convective LH from

GOES-16 are very similar to those from CSH that uses model simulations in the LUT. Their vertical profiles show heating throughout the vertical layer except near the surface where evaporation occurs, and heating peaks around the middle of the atmosphere. This vertical pattern differs from when using an empirical formula with radar reflectivity. Vertical profiles of LH from NEXRAD highly depend on vertical profiles of reflectivity which typically peaks near surface in convective regions, and thus, maximum LH is usually observed at lower level, which is not commonly shown in the modeled heating rate. Even though

their vertical shape is slightly different, the total LH over convective clouds is shown to be similar. Similar magnitude of LH between GOES-16 and NEXRAD suggests a potential use of LH from GOES-16 in initiating convection in the regions where ground-based radar data are not available.

**Acknowledgments**

This research is supported by the Cooperative Institute for Research in the Atmosphere (CIRA)'s Graduate Student Support Program.

**Author contributions**

All three authors contributed to the retrieval, and the manuscript was written jointly by YL, CK, and MZ.

**Competing interests**

The authors declare that they have no conflicts of interests.

**Data availability**

GOES-16 data are obtained from CIRA, but access to the data is limited to CIRA employees. GPM DPR data are from: GPM DPR and GMI Combined Convective Stratiform Heating L3 1 month 0.5 degree x 0.5 degree V06, Greenbelt, MD, USA, Goddard Earth Sciences Data and Information Services Center (GES DISC), Accessed: **[April 5th,**

**2021]**, 10.5067/GPM/DPRGMI/CSH/3B-MONTH/06, GPM DPR Spectral Latent Heating Profiles L3 1 month 0.5 degree x 0.5 degree V06, Greenbelt, MD, USA, Goddard Earth Sciences Data and Information Services Center (GES DISC), Accessed: **[*April 5th, 2021*]**, 10.5067/GPM/DPR/SLH/3A-MONTH/06, and GPM DPR and GMI Combined Stratiform Heating L2 1.5 hours 5 km V06, Greenbelt, MD, USA, Goddard Earth Sciences Data and Information Services Center (GES DISC), Accessed: **[*April 5th, 2021*]**, 10.5067/GPM/DPRGMI/CSH/2H/06.




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
