# Peer review of "Latent heating profiles from GOES-16 and its comparison to heating from NEXRAD and GPM"

_Atmospheric Measurement Techniques, 2021_

## Referee Comment (RC2)

Latent heat (LH) released during the atmospheric convection is a very important parameter as it creates the buoyancy that further promotes the development and intensification of convection and the up-scale growth of the convective systems.  Indeed, the LH associated with the tropical convection is a major driver of the tropical large-scale circulation.

Unfortunately, there are no direct measures of LH.  Several algorithms have been developed in the past to estimate the LH from microwave (passive and active) satellite observations of precipitation.  This paper is innovative as it proposes a new way for the estimation of LH from the GOES-16 ABI geostationary observations.

While the geostationary observations offer much better spatial and temporal resolution their main disadvantage is that they offer observations of the cloud tops only with little information on the precipitation structure below the cloud deck.  In that sense, the only observations that would be useful for the estimation of the latent heat would be that of active convection that leaves an imprint on the cloud deck.  The authors acknowledge that and focus on estimating the LH of convective events.  In that sense, the study is driven by the desire to provide LH estimates to improve the initialization of high-resolution models, similar to the path that has been used with ground-based radar observations.  And this is an important motivation.  However, I would strongly suggest modification of the title which currently implies that LH can be retrieved for all types of precipitation.

Furthermore, even with regard to the convective events only, it should be pointed out that the LH cannot be retrieved for all convective storms. It could be retrieved for only two types of convective events - the strong convective regions that would project onto the cloud deck of large systems; the initial convection that develops in otherwise clear regions.  So convective events that develop under cloud shields but are not strong enough or deep enough to project onto the cloud tops would not be included in the retrievals.

The paper addresses an important area for future investigations and provides an innovative approach. However, the paper does not provide sufficient detail on the methods that are used. And it does not discuss how the difference in the LH estimations from different instruments and methods would impact the intended application – providing LH estimates for initialization of convection in high-resolution models.

Considering all this I propose the paper be evaluated again after a major revision.

More specific comments:
1. On the method:
    a. The identification of the convection is based on two criteria: i) detection of the overshooting tops using thresholds that are applied to the TBs; ii) detecting "Bubbly" cloud deck.  There are several problems with that:
       i. The definitions of these two characteristics of the cloud deck are quite vague.
       ii. The paper does not provide any quantitative thresholds or a clear description of an algorithm. Lines 228-238 provide a reference and a very high-level description of an

approach but without a sufficient detail.  This description includes the term "lumpy cloud tops" but how are these "Bubbly clouds" identified??  How has the algorithm for convection detecting that is used here been evaluated?  Or has it been evaluated?  How does it compare to other algorithms of this kind?

   iii.  The paper does not discuss how the convection detecting algorithm performs for different types of convective clouds – e.g. how does it perform for shallow and isolated versus its performance for deep and organized clouds.

   iv.  The detection of convection from GOES-16 ABI observations is a central piece of the proposed new method (the other central piece being the construction of the LUT tables). So, a much more detailed description is needed.

  b.  The creation of the LUT tables

   i.  Lines 266-269 – it is not clear how the observations are clustered in the LUT table.  The text says: "*In each scene, convective grid points are defined by the threshold found in the previous section, and neighboring convective grid points are clustered to form a convective cloud. Minimum Tb of each cloud is calculated, and LH profiles from the clouds with the same minimum Tb are averaged to produce mean profiles for each Tb bin of the LUT.*"  Some questions:

     1.  which thresholds? On vertical velocity?

     2.  *neighboring convective grid points* – how many neighbors?

     3.  *Minimum Tb of each cloud is calculated* – I thought that the Tb calculations from the model are not reliable.  Did I get this wrong?

   ii.  On the use of LUT tables:

     1.  It will be useful to discuss that the LH retrievals from different methods / instrument all use different LUT tables.

     2.  It will be very good if these LUT tables are based on the same model simulations.  This is probably not feasible for this study but it should be discussed and maybe future research should address this.

     3.  Also, future research may use the same model simulations and compute the parameters that are used by the different techniques.  After that it would be possible to investigate how the features that are identified by the different techniques compare (map) to each other – e.g. how do the features defined by the NEXRAD criteria compare to these defined by the, say, CSH criteria, or to those defined by theGOES-16 ABI criteria.  This is really a big project that stands on its own.  I am not suggesting the authors do this here but it would be very beneficial if they discuss the need for these analyses.

2.  On the results:

  a.  Applying the three different methods for LH estimation based on NEXRAD, CSH and GOES-16 observations over the same scene illustrates the similarities and the differences between the approaches.  While the magnitudes of the estimated heating may be comparable among the three approach, there are distinct difference in the vertical structure of the estimated LH, with the NEXRAD estimations being the most different from the other two.  There are two important things to note:

   i.  NEXRAD estimations show significant variability in the individual columns – both in intensity and in vertical structure.  The CSH estimations show less variability and the

GOES LH estimates show the least variability, with an almost constant height of the peak (Figs 5 and 6 show this most clearly).

   ii. The vertical structure of the NEXRAD profiles is distinctly different from the other two – with having a peak at a lower altitude.

b. While the above features are discussed in the paper, there is no discussion on how these differences would likely affect the intended application of the GOES LH retrievals. Would the GOES LH retrievals be useful for initialization of convection in high-resolution models in the same way in which the NEXRAD LH retrievals have been?  This is a very important implication!  Ideally this could be addresses by applying the three LH estimates in the initialization of a high-resolution model and comparing the impacts on the convective forecast.  This would be the ultimate, and a very interesting test! Alternatively, these implications should be at least discussed in more detail.

In summary: proposed are the following modifications
- Change the title
- discuss in more detail the limitations of the approach – only possible for convective events and then only for some types of these events – deep and strong convection that leaves imprints on the cloud deck; isolated convection
- provide a detailed description of the algorithms that were used and how they perform for different convective cloud types.
- Discuss the implications of the differences in the LH profiles that are retrieved by the different algorithm using the same observations
- Provide a discussion on the way forward